# Evolution of Plant Na^+^-P-Type ATPases: From Saline Environments to Land Colonization

**DOI:** 10.3390/plants10020221

**Published:** 2021-01-24

**Authors:** Siarhei A. Dabravolski, Stanislav V. Isayenkov

**Affiliations:** 1Department of Clinical Diagnostics, Vitebsk State Academy of Veterinary Medicine [UO VGAVM], 21002 Vitebsk, Belarus; sergedobrowolski@gmail.com; 2International Research Centre for Environmental Membrane Biology, Foshan University, Foshan 528000, China; 3Department of Plant Food Products and Biofortification, Institute of Food Biotechnology and Genomics NAS of Ukraine, 04123 Kyiv, Ukraine

**Keywords:** salinity, P-type ATPases, molecular evolution, domain architecture, Na^+^/K^+^-P-ATPases, phylogeny

## Abstract

Soil salinity is one of the major factors obstructing the growth and development of agricultural crops. Eukaryotes have two main transport systems involved in active Na^+^ removal: cation/H^+^ antiporters and Na^+^-P-type ATPases. Key transport proteins, Na^+^/K^+^-P-ATPases, are widely distributed among the different taxa families of pumps which are responsible for keeping cytosolic Na^+^ concentrations below toxic levels. Na^+^/K^+^-P-ATPases are considered to be absent in flowering plants. The data presented here are a complete inventory of P-type Na^+^/K^+^-P-ATPases in the major branches of the plant kingdom. We also attempt to elucidate the evolution of these important membrane pumps in plants in comparison with other organisms. We were able to observe the gradual replacement of the Na+-binding site to the Ca^2+^-binding site, starting with cyanobacteria and moving to modern land plants. Our results show that the α-subunit likely evolved from one common ancestor to bacteria, fungi, plants, and mammals, whereas the β-subunit did not evolve in green algae. In conclusion, our results strongly suggest the significant differences in the domain architecture and subunit composition of plant Na^+^/K^+^-P-ATPases depending on plant taxa and the salinity of the environment. The obtained data clarified and broadened the current views on the evolution of Na^+^/K^+^-P-ATPases. The results of this work would be helpful for further research on P-type ATPase functionality and physiological roles.

## 1. Introduction

Land soil salinity is a growing problem worldwide. For example, the reduction in crop productivity because of an increase in salinity has caused economic losses of over 31 billion USD annually [1,2]. Soil salinization takes up to 1.5 million ha of farmland per year out of production and decreases the production potential of up to 46 million ha per year [2]. Soil salinization tends to increase by 10% per year worldwide [3,4].

Na^+^ and Cl^−^ ions are the most abundant ions in saline environments and are considered to be the main cause of ion toxicity caused by salinity. Plants can accumulate Na^+^ and Cl^−^ in sufficient amounts to adjust the osmotic pressure; however, an excess of these ions in cells and plant tissues could cause toxic effects. To maintain active metabolic activity and plant growth, plant cells need the presence of K^+^ in substantial concentrations. K^+^ uptake from the soil is mediated by several ion transport systems driven by the electrochemical gradient of the plasma membrane of the cells. K^+^ and Na^+^ are by far the most abundant monovalent inorganic cations, with very similar physico-chemical characteristics. These two cations are considered to be the main competitors in cellular transport [5]. Therefore, K^+^ and other types of transporters can mediate Na^+^ leaks caused by high salinity and support the unavoidable constant influx of Na^+^ [6,7,8,9]. Salt-stressed plants need to maintain cytosolic Na^+^ (Cl^−^) concentrations at levels that are nontoxic to their cells [5,10].

Eukaryotes recruit two main transport systems for the active Na^+^ efflux: cation/H^+^ antiporters (CPAs) and Na^+^-P-ATPases [10,11]. The CPA antiporters of the plasma membrane are considered to be the primary Na^+^ efflux systems in plant cells [12,13]. Only two transport proteins exhibiting Na^+^(Li^+^)/H^+^ antiport activities were identified in plants—namely, AtNHX7 (AtSOS1) and AtNHX8 [10,14]. Importantly, many reports suggest that the overexpression or heterologous expression of SOS1-like genes in plants leads to increased salt tolerance [15,16,17,18,19].

P-type ATPases are a large family of primary transporters powered by the hydrolysis of ATP [20,21]. According to their transport properties and domain structure, the ATPases are divided into five major subfamilies (pp.1–5, [22,23]). These transport proteins are involved in the transport, redistribution, and removal of metals such as Zn and Cu (Zn^2+^- and Cu^2+^-ATPases, PIB group), intracellular Ca^2+^ signaling, cellular compartmentalization (Ca^2+^/Mn^2+^-ATPases, P2A-B groups), the generation of a membrane electrochemical gradient to supply driving force for the secondary transporters including Na^+^/H^+^ CPA exchangers (H^+^-ATPases, P3 subfamily) and membrane vesicle trafficking and budding (phospholipid-ATPases, P4 subfamily), and Na^+^ and K^+^ transport and removal (Na^+^/K^+^-ATPases, P2C-D groups) [23]. Although there is some evidence that members of P5 subfamily ATPases are involved in vesicle budding from the endoplasmic reticulum, the exact role of this type of P-ATPases remains unknown [24]. However, the P5 ATPases are likely to belong to transmembrane helix dislocases [25,26].

The Na^+^-P-ATPases are divided between the animal cell-specific Na^+^/K^+^-P-ATPases and the fungal Na^+^-P-ATPases [27]. It is thought that flowering plants lack Na^+^-P-ATPases [11,28,29]. Literature data suggest that proton gradients are maintained by plasma membrane H^+^-ATPases in plants and fungi, whereas animal cells use Na^+^ and K^+^ gradients by recruiting Na^+^/K^+^-P-ATPases. However, the question regarding the possibility of Na^+^-P-ATPases’ presence in some flowering plants or the involvement of some plant P-ATPases in Na^+^(K^+^) transport remains open. Due to the widely accepted model that flowering plants do not express Na^+^-P-ATPases, the theory of evolution of flowering plants in non-saline oligotrophic environments with a low Na^+^ content has a very strong point [30]. The low Na^+^ concentration in the environment is not necessary for the system to mediate an extensive Na^+^ efflux point [30]. Additionally, in low-Na^+^ environments, it is impossible to generate a steep electrochemical gradient to energize the plasma membrane using Na^+^ [28]. However, all the Na^+^-P-ATPases are present in some primitive plants [11,23,31]. Interestingly, it was found that moss *Physcomitrella patens* (*Bryopsida*) have three P-ATPases similar to the fungal ENA-Na^+^ [28,29,30,32,33]. Thus, the presence of the Na^+^-P-ATPases in moss and Na^+^/K^+^-P-ATPases in green algae indicates that these enzymes were likely lost during early land colonization by flowering plants [30]. However, the exact physiological role of *P. patens* ENA-Na^+^-P-ATPases remains unclear. It was shown that ENA Na^+^-P-ATPases from *P. patens* function cooperatively with the SOS1-like CPA transporters, whereas ENA-type Na^+^-P-ATPases in fungi are considered to be the major Na^+^ exporters [11,34]. Noteworthy, it was reported that wild-type *P. patens* plants grow slower than an ENA1 mutant [33]. Moreover, the heterologous expression of ENA1 from *P. patens* in rice could enhance salinity tolerance; however, in normal conditions ENA1-expressing plants were less fit than control lines [35]. The expression of a yeast ENA1 homolog in tobacco-cultured cells and *Arabidopsis thaliana* leads to an increase in salinity tolerance, however the unselective export of K^+^ as well as Na^+^ was detected in transgenic tobacco cells [36,37]. Thus, the maintenance of Na^+^-P-ATPase activity in non-saline conditions would be difficult to sustain in low-Na^+^ conditions. Moreover, the unselective transport properties of yeast ENA1 to transport both K^+^ and Na^+^ would be not desirable to improve plant growth and salt tolerance [37].

The activities of Na^+^-P-ATPase have been identified for various algal species, including *Tetraselmis viridis* (Prasinophyceae) [38,39] and *Dunaliella parva* [40]. There was an early identification of the Na^+^-ATPase gene from *Heterosigma akashiwo* [41,42] and the more recent discovery of genes encoding Na^+^-ATPases in red and green algae *Chondrus crispus*, *Chlamydomonas reinhardtii*, *Ostreococcus tauri*, and others are very similar to animal Na^+^/K^+^-P-ATPases [31,43,44,45,46]. The EST (Expressed Sequence Tags) analysis of *Porphyra yezoensis* gametophytes and sporophytes revealed the presence of Na^+^/K^+^-P-ATPase (PyKPA1) similar to that in animals [47]. Further functional analysis was conducted using the heterologous expression of a full-length clone in *Escherichia coli* and yeasts [48]. The expression of PyKPA1 in rice led to the elevation of salinity tolerance in transgenic plants [49]. Thus, it could be summarized that the identified algal Na^+^-P-ATPases belong to the animal type of Na^+^/K^+^-P-ATPases, whereas the Na^+^-P-ATPases discovered in mosses are likely to have similar roots to the fungal ENA type of Na^+^-P-ATPases.

The structure of Na^+^/K^+^-P-ATPase is complex. Close homologues of animal Na^+^/K^+^-P-ATPases have been identified in some archaea, red and green algae, and oomycetes [31,48]. The “classical” structure of Na^+^/K^+^-P-ATPase comprises three different subunits: α, β, and γ-subunits. Usually, Na^+^/K^+^-P-ATPase is a binary polypeptide complex comprising the main catalytic α-subunit and smaller glycosylated β-subunit [49]. The third known small γ-subunit belongs to the FXYD family and is involved in the regulation of pumping activity. However, the γ-subunit was identified exclusively in metazoan and does not occur in other taxa including plants [50]. The β-subunit is involved in the stabilization of the α-subunit and the occlusion of the K^+^ ions, whereas the third subunit γ regulates pumping activity in the tissue [51,52]. In contrast with animal Na^+^/K^+^-P-ATPases, the P-type Na^+^ pumps in red and green algae lack “typical” highly glycosylated β-subunits [53]. Predicted conserved insert exposed to the extracellular side between M7 and M8 in some algal Na^+^/K^+^-P-ATPases is suggested to be the potential equivalent of the β-subunit of the animal P-type Na^+^ pumps. However, this hypothesis does not have any experimental support [11].

Despite the proposed model that flowering plants have lost their Na^+^-P-ATPases during evolution and land colonization, some reports suggest the existence of putative ouabain-sensitive Na^+^/K^+^-P-ATPases in flowering plants [54,55]. However, these data have to be confirmed by the identification and functional characterization of these predicted transport proteins. Thus, the absence of Na^+^/K^+^-P-ATPases in flowering plants opens a range of questions that need to be answered. In which direction has the evolution of plant Na^+^-P-ATPases moved? How strict is plant ATPases’ substrate specificity? How are the properties of plant ATPases changed during movement from the saline environment to the terrestrial way of life?

This study aimed to provide a structural and phylogenetic analysis of plant Na^+^/K^+^-P-ATPases and consider the possible mechanisms of P-ATPases evolution from unicellular systems of red and green algae to the complexity of flowering plants.

## 2. Results

### 2.1. Domain Organization of the Plants P-ATPases, Comparison to Animals

Domain organization plays a pivotal role in understanding the functional properties of proteins and their evolutionary course. Therefore, to gain further insight we have inspected the domains in each clade and have found their association with functionally divergent residues. In this work, we have identified the P-ATPase domain organization of the model plant *Arabidopsis thaliana* and species from other taxa (Figure 1) (Appendix A). In total, we have defined five families of P-type ATPases in Arabidopsis: P1B, P2A, and B, P3A, P4, and P5. It was found that three domains are common for each family/subfamily: E1-E2 ATPase (PF00122), haloacid dehalogenase-like hydrolase (PF00702), cation transport ATPase (P-type) (PF13246). Each family/subfamily of P-type ATPases comprises a unique domain, distinguishing it from the other groups. According to our data, the P1B-type heavy metal cation-transporting ATPase family has a heavy metal-associated domain (PF00403). P2B-type calcium cation-transporting ATPases have both cation transporter/ATPase, N-terminus (PF00690), and cation transporting ATPase, C-terminus (PF00689). There are two conserved N-terminal (PF00690) and C-terminal (PF00689) domains found in several classes of cation-transporting P-type ATPases, including those that transport H^+^, Na^+^, Ca^2+^, Na^+^/K^+^, and H^+^/K^+^. As the main difference, the N-terminal domain could undergo reversible sequential phosphorylation and conformational changes that may be required for its regulation. Interestingly, our analysis revealed that proteins from the P2A type (subgroup 2: At4g00900) and P2B type (subgroup 1: At3g63380, At3g22910, At5g53010) have the same domain architecture (Figure 1 and Figure 2). The P2B-type group 2 has an important Ca^2+^-ATPase N terminal (N-terminus?) autoinhibitory domain (PF12515). In contrast, the P3A-type proton-translocating ATPase family exhibits the presence of only cation transporter/ATPase, N-terminus (PF00690), without cation transporting ATPase, C-terminus (PF00689). The analysis of the P4-type family (catalytic ATPase component of the phospholipid flippase complex) revealed the existence of two unique domains: phospholipid-translocating ATPase N-terminal (PF16209) and phospholipid-translocating P-type ATPase C-terminal (PF16212). The P5-type family is represented only by E1-E2 ATPase and overlapping domains (haloacid dehalogenase-like hydrolase and cation transport ATPase (P-type).

Thus, the domain architecture of all P-type ATPases from the studied species was identified (Appendix A). Our analysis confirmed that all the known sodium/potassium-transporting ATPases (further as Na^+^/K^+^ -P-ATPases) from animals and red and green algae belong to the P2B type (subgroup 1, or P2A subgroup 2).

#### 2.1.1. Comparison of the Ca^2+^ and Na^+^/K^+^ Binding Sites

The crystal structure of two animal Na^+^/K^+^-P-ATPases was reported. The sites/amino acids required for the ion’s coordination (amino-acid residues that have an electron donor (O, N, S, and Cl) within 3 A^o^ of the metal ion are considered to be coordinating residues) were identified. We conducted a comparison of Ca^2+^ and Na^+^ (depicted in magenta circles) binding sites from known sodium/potassium-transporting P-ATPases (identified by [51,56], respectively). Additionally, Na^+^ binding sites have been identified in *Sus scrofa* (4HQJ) [57] (marked with green squares) (Appendix A).

According to our analysis, many coordinating residues are conserved between red and green algae, Arabidopsis, and animals. Only two proteins from Arabidopsis (calcium-transporting ATPase 2 and 3) have all Ca^2+^-coordinating residues identical to rabbit SERCA1 (Appendix A).

Further analysis of the Na^+^-coordination sites in P-ATPases reveals the high complexity of these structures in different taxa. The ideal match for the required residues was found only for the animals, while red and green algae and Arabidopsis have some deviation, which occurred mainly within the first residues. Interestingly, in comparison with other taxa, the transmembrane segment M7 coordinating residues (I and Q) are completely missing in Arabidopsis (Appendix A).

#### 2.1.2. Phylogeny of the α-Subunit

To evaluate the phylogeny of the P2B-type ATPases, we collected sequences belonging to this type of ATPases from different taxa (bacteria, red and green algae, fungi, animals (α-subunit), listed in Appendix A) (Figure 3). Our analysis revealed the distribution of P2B P-type ATPases among five different clades. The first clade includes Ca^2+^-transporting ATPases from mono- and dicotyledons, clubmosses, moss, and spikemosses (sub-clade Ia). Sub-clade Ib, on the other hand, combines more primitive taxa: yeast and red and green algae.

All the identified Na^+^/K^+^-P-ATPases have been located to clade II. Interestingly, this clade includes evolutionally very distant species: animals and red and green algae. Bacterial magnesium-transporting ATP (*Escherichia coli* P0ABB8) and liverwort cation ATPase (*Marchantia polymorpha A0A176WQH1*) have served as parental forms.

Archaeal, green photosynthetic bacteria (cyanobacteria, chloroflexi, and chlorobi), and yeast have been grouped to the IIIa sub-clade. The IIIb sub-clade includes Verrucomicrobia (*Phragmitibacter flavus* A0A5R8KH12), Chloroflexi (*Chloroflexus islandicus* A0A178MNY7), and green algae (*Chlamydomonas eustigma* A0A250 × 2T5). Thus, the III clade comprises P2B P-type ATPases from the most primitive organisms.

Clade IV combines different cation P-ATPases and Ca^2+^-transporting P-ATPases from diverse taxa. We could distinguish IVa sub-clade from the IVb sub-clades by the presence of the red algae proteins: Porphyridium purpureum (A0A5J4Z959 and A0A5J4Z564).

Yeast Na^+^/K^+^-P-ATPases were located to the Vth clade altogether with putative moss Na^+^/K^+^-P-ATPases (Physcomitrella patens Q7XB51, Q7XB50, C1L359) and cation P-ATPases from liverwort (Marchantia polymorpha A0A176WIA1, A0A176W455, A0A176WCE3).

#### 2.1.3. Phylogeny of the β-Subunit

Previous studies have reported that plants do not have a β-subunit, which has been shown to be required to stabilize Na^+^/K^+^ transport properties. Interestingly, we have found β-subunit only in one plant species, *Anthurium amnicola* (Dressler) (A0A1D1Y1U2), and in one bacterium—*Gammaproteobacteria bacterium* 2W06 (PYZ99283.1). According to our analysis, those β-subunits have a very high level of similarity with the β-subunit from human/metazoa proteins (Appendix A). In addition to animals, the β-subunit is also widely presented in insects.

We have identified two insertions in the C-terminal insertion of some red and green algaes’ α-subunit with a high similarity to the animal’s β-subunit (Appendix A). As we could see, the first insertion is presented only in red algae (A0A5J4YS69: *Porphyridium purpureum*; A0A2V3IH88, *Gracilariopsis chorda*) (Appendix A). Previously, Pedersen et al. [11] also found an β-subunit-like insertion (in our case, it is the second insertion (Appendix A)). Interestingly, all red and green algae insertions have different lengths (approximately 100–150 aa, compared with approximately 300aa for the β-subunit length in animals and insects). Moreover, *Ostreococcus lucimarinus* does not have such an insertion, whereas *Porphyridium purpureum* and *Gracilariopsis chorda* have the shortest length of insertion. Therefore, these insertions are closest to each other sequences on the phylogenetic tree (Figure 3). Surprisingly, we have also discovered β-subunit-like proteins in several amoeba viruses (Mimivirus genus), with sequences more similar in length and amino acids to the red algae’s insertion. Consequently, those virus proteins were used in our phylogenetic tree as an outgroup. The resulting tree (Figure 4) suggests the closest relations are between the virus and red algae insertion, while green and red algae proteins were located on separate branches. Similarly, Metazoa proteins have been separated and not much related to bacterial, plant, or Insecta proteins.

#### 2.1.4. Comparison of the Binding-Sites

In the recent paper, a set of amino acids important for the interaction between α, β, and γ-subunits, were identified in *Sus scrofa* (4HQJ) [57]. The α-subunit has three regions which are required for proper interaction with the β-subunit (Appendix A). As we could see, the majority of the required sites in flowering plants and red and green algae are missing. However, all the studied α-subunits have a conserved catalytic site (Appendix A).

Furthermore, flowering plants and red and green algae do not have sites required for interaction with the γ-subunit (Appendix A). Red algae (*Porphyridium purpureum* and *Gracilariopsis chorda*), on the contrary, exhibit the highest similarity level to the animal α-subunits and closest match to the required sites.

Algae sequences (insertions, extracted from theα subunit) have been aligned with β-subunits from metazoan, plant *Anthurium amnicola* (A0A1D1Y1U2), and bacterium *Gammaproteobacteria bacterium 2W06* (PYZ99283.1) (Appendix A). Similar to the other interaction sites, insertions are missing almost in all the required sites. On the contrary, the plant and bacterial sequences are very similar to the metazoan and exhibit the presence of almost all sites. Thus, our analysis further supports our previous observation of their close phylogenetic relation.

## 3. Discussion

The main component of Na^+^-P-ATPases is the α-subunit. It is well known that the α-subunit plays the main role during the transport of the three cytoplasmic Na+ in exchange for two extracellular K^+^ through alternating E1/E1P and E2P/E2 states. It has been suggested that the β-subunit is required for the stabilization of the α-subunit during conformational rearrangements of the E2-to-E1 transition [58]. In particular, the γ-subunit interferes with Na^+^ release during conformational changes between the stable IIIb and from the transient site IIIa [56,57,58,59]. However, it was found by Nyblom et al. [57] that the β- and γ-subunits do not provide a substantial influence on the structure. For this reason, we could speculate the that β- and γ-subunits are required only for animal cells.

On the other hand, plant cells have a different structure—a solid cell wall, strengthened by the lignins, suberins, and other compounds—suggesting that β- and γ-subunits are not necessary. Bacterial cells have several members of the P-type ATPases but, similarly to plant cells, do not have β- and γ-subunits (Figure 5). However, no direct experimental evidence is known so far to support this suggested model.

It was proposed that land colonization and the shift from salt sea to fresh in-land water were the main events triggering the evolution and gradual conversion of the Na^+^/K^+^-P-ATPase to the Ca^2+^ transporting P-type ATPases [11]. The results of our analyses support this hypothesis. We also could see several intermediate stages of this transition in the sequences of green and red algae (Figure 5). Particularly, β-subunit-like insertions into the α-subunit are present mostly in sea green and red algal species, while freshwater species exhibit an absence of such an insertion. The only exception that we have found is the α-subunit of P-type ATPase with β-subunit-like insertions of the freshwater green algae *Micractinium conductrix*. The concentration of salt in the environment likely was one of the key factors in changing the P-type ATPases substrate specificity. Up to now, the exact functional role of this insertion remains unknown. However, it is obvious that these β-subunit-like insertions are present only in a small fraction of algal species.

The second important factor of structural differences between animal and plant P-type ATPases is the appearance and improvement of the cellular walls. In this case, we also could see a sufficient difference in evolution between green and red algae. Interestingly, from the green algae we could trace the rise and spread of the several gene families which are required for the formation of the cellular wall and primarily involved in lignin and suberin metabolism. For example, (+) and (−)-pinoresinol-forming dirigent genes, known from bacteria to flowering plants [60], play a significant role in the lignan/lignin metabolism and have been found only in the green algae [61]. However, lignin-like compounds have been identified in both red [62] and green algae [63].

Interestingly, some experimental data suggest the existence of dual-transport activity (Na^+^/K^+^ and Ca^2+^) for members of P-ATPases in Mycobacteria [64,65]. Additionally, experiments with chimeric mammalian Ca^2+^-ATPase/Na^+^/K^+^-ATPase molecules demonstrate that the central cytoplasmic domain of the Ca^2+^-ATPase does not have primary importance for the binding and occlusion of Ca^2+^, whereas it plays an important role in the ElP->-E2P conformational changes [66]. Although some experimental data demonstrate that the localized-to-small-vacuoles AtACA4 and ER-specific AtACA2 from Arabidopsis thaliana are involved in NaCl tolerance [67,68], there has been no direct experimental evidence of Na^+^/K^+^ or Na^+^ activities for plant Ca^2+^-P-ATPase reported yet. Nevertheless, we cannot omit the possibility that some Ca^2+^-P-ATPases of flowering plants could possess Na^+^/K^+^ transport properties.

A comparison of the Ca^2+^ and Na^+^-binding amino acids from known Na^+^/K^+^-P-ATPases with solved crystal structures [51,54,57] is presented on Appendix A. We could notice that the Ca^2+^-binding sites are conserved during the evolution. In every taxon, we could see the identical Ca^2+^-binding sites or a very high level of similarity of these sites with the defined standard. Interestingly, bacteria, metazoa, green and red algae, and flowering plants share basically the same or nearly the same sites (AIENENTDE).

The analysis of the Na^+^-binding sites revealed a completely different situation. The precise sequence of the binding sites could be found only in animals. Some high-similarity proteins could be found in green and red algae (*Ostreococcus lucimarinus*, A4RQL0, and *Gracilariopsis chorda*, A0A2V3IH88, respectively), while other sequences have a high degree of variation throughout studied taxa. A recent paper suggests that specificity to the particular ion is defined by the physical properties of the P-type ATPase protein. This specificity could be shifted towards another ion as a result of a natural mutation or artificially rewired [69]. Altogether, in addition to our bioinformatic analyses (Appendix A), we can propose that the ion specificity of the original Na^+^/K^+^-P-ATPases was gradually shifted during the green lineage evolution to the Ca^2+^ transporting P-type ATPases. During this process, Na^+^/K^+^ binding sites were lost and/or replaced by the Ca^2+^ binding residues (Appendix A).

Most probably, the green lineage of P-type ATPases does not follow the common pathway of the vertical inheritance or symbiotic gene transfer. This suggestion is supported by our phylogenetic tree, where prokaryotic organisms are separated from eukaryotic (Figure 3 and Appendix A). We could not trace the expected connection between cyanobacteria, green algae, and flowering plants. On the contrary, photosynthetic bacteria have isolated branches (IIIa and IIIb), with archaea members as the closest match in the case of IIIa. In contrast with photosynthetic bacteria, the green algae are represented in almost every clade (except Vth).

Thus, as a general direction of the plant evolution and adaptation related to the Na^+^/K^+^-P-ATPases, we can propose the following hypotheses: (1) the further presence of the Na^+^/K^+^-P-ATPase was not needed after land colonization and due to the low Na^+^ concentration in the environment; (2) the function of the Na^+^/K^+^ transport was switched to other, more specialized/specific and efficient—in terms of energy consumption—transporters, including CPA I and II; (3) architecture of the roots with a line of physical defense mechanisms (strengthening cell-wall, incorporating into the cellular wall lignins and suberins) that do not match the mechanics of the Na^+^/K^+^-P-ATPases.

According to our phylogenetic analysis of the α-subunits of red algae, the structure of Na^+^/K^+^-P-ATPase shares the most similarities with corresponding animal proteins. The situation gets even more confusing when it is followed by green algae proteins. As we have noticed, some algae (red and green) have β-subunit-like insertions in the α-subunit. Both insertions (1st and 2nd) have a substantial length (combined up to 150 aa). It would be expected that α-subunits with β-subunit-like insertion/s would have a lower general sequence similarity with all other α-subunits. Such lower similarity should have resulted in the cluster on the far separate branch. However, our analyses exhibit a completely different picture. Therefore, this question requires further elaboration.

The origin and nature of the insertion is another interesting question. We could suggest several possibilities:
(1)Ancient transduction. We found nine β-subunit-like proteins in several amoeba viruses (Mimivirus genus). While amoeba is the current target for the virus, it is possible, that earlier in evolution virus was able to facilitate the transfer of some genetic material to the algae. Interestingly, this idea is not a novelty for red algae. It was shown that the *Ectocarpus siliculosusvirus* (EsV-1) pandemic for the marine brown algae, *E. siliculosus*, encodes a protein similar to the histidine kinase. Histidine kinase is a crucial part of the two-component signaling pathway, common for bacteria, fungi, and algae. In plants, histidine kinases participate in the signaling pathway of the plant hormone cytokinin, which regulates multiple vital processes [70].(2)The acquisition of the β-subunit-like protein with transposable elements TEs. It is known, that red algae have passed two phases of large-scale genome reduction [71] and have incorporated multiple TEs in their genome with the subsequent recombination of the TEs, which could provide an evolutionary advantage [72].(3)The preservation of genes after horizontal and endosymbiotic gene transfer (HGT and EGT). Previous research data suggest that red algae stand between prokaryotes and green algae, acting as a mediator in horizontal and endosymbiotic gene transfer [73]. The study of the red alga *Porphyridium purpureum* genome suggests that red algae have saved many genes that green algae lost during evolution [74].

We cannot omit the possibility that some Ca^2+^-P-ATPases of flowering plants could possess Na^+^/K^+^ transport properties. Some results of our bioinformatics analyses suggest a possibility for a gradual transition from the Na^+^/K^+^ to the Ca^2+^ transporting P-type ATPases, followed by the loss and/or replacement of the Na^+^/K^+^ binding sites by the Ca^2+^ coordinating residues (Appendix A). Interestingly, some experimental data suggest the existence of dual transport activity (Na^+^/K^+^ and Ca^2+^) for members of P-ATPases in Mycobacteria [64,65,74]. Additionally, experiments with chimeric mammalian Ca^2+^-P-ATPase/Na^+^/K^+^-P-ATPase molecules demonstrate that the central cytoplasmic domain of the Ca^2+^-ATPase does not have primary importance for the binding and occlusion of Ca2+, whereas it plays an important role in the ElP-->-E2P conformational changes [66]. Although some experimental data demonstrate that the localized to small vacuoles AtACA4 and ER specific AtACA2 from *Arabidopsis thaliana* are involved in NaCl tolerance [67,68], there is no direct experimental evidence of Na^+^/K^+^ or Na^+^ activities for plant Ca^2+^-P-ATPase reported yet. Interestingly, the prokaryotic lineage is separated from the eukaryotic (Figure 3 and Figure 4). We could not track the expected connection between cyanobacteria, green algae, and flowering plants. On the contrary, photosynthetic bacteria have isolated branches (IIIa and IIIb), with archaea members as the closest match in the case of the IIIa. In contrast with photosynthetic bacteria, the green algae are represented in almost every clade (except Vth). Surprisingly, one bacterial magnesium-transporting ATP (*Escherichia coli* P0ABB8) was located to the Na^+^/K^+^-P-ATPases (II clade). Thus, the results of our phylogenetic study suggest the highly adaptable nature of the P-type ATPases.

New data from the genome sequencing and transcript profiling of new plant species will become available in the near future for the wide scientific community. It would be an important task to conduct careful result analysis of these studies and scan for the new candidate proteins exhibiting Na^+^/K^+^- or Na^+^-transport properties. The deep understanding of the molecular evolution of Na^+^/K^+^-P-ATPases will provide new insight that will substantially expand our knowledge of the biological functions and properties of these membrane proteins. The data presented and analyzed in this study contribute to our current understanding of plant P-ATPases evolution and the structure–function relationship of these proteins. Thus, re-engineering of P-type ATPase to prevent H^+^ or Na^+^ leaks from them [23] or another modification of transport properties, tissue, and cellular localization, as well as stress inducibility, are the main directions to improve the efficiency and organism suitability of these transport proteins. The appearance of genome editing tools offers a wide range of possibilities to improve the Na^+^/K^+^- transport or other properties of P-type ATPases as well as plant tolerance in general.

## 4. Materials and Methods

### 4.1. Sequences Retrieval

The sequences of 48 Arabidopsis P-type ATPases and α, β, and γ-subunits of the animal Na^+^, K^+^-ATPases were extracted from Uniprot database and used for the following BLAST [75] searches in NCBI, InterPro [76], and Pfam [77] databases to identify related sequences in other species (Appendix A). All partial and fragmented sequences were eliminated. Na^+^ and K^+^-ATPase with known structures (Sus scrofa 4HQJ (10.2210/pdb4HQJ/pdb) [57]) were downloaded from the PDB database (http://www.rcsb.org/) [78] and used as a baseline for a structure comparison (α-4HQJ_1, β-4HQJ_2, and γ-4HQJ_3 subunits).

### 4.2. Domain Organization and Active Sites

The domain architecture of the P-ATPases was defined with CD-search (NCBI) [79] and MOTIF search (KEGG) [80] tools with E-value (*p* ≤ 0.001). The further classification was based on the P-ATPases from the *Arabidopsis thaliana* (Figure 1). Residues contributing with Ca^2+^ and Na^+^ coordination were highlighted according to [56] and [51], respectively. PSIPRED server (http://bioinf.cs.ucl.ac.uk/psipred/) was used to predict proteins’ topology, transmembrane domains, and signaling peptides [81].

### 4.3. Multiple Sequence Alignments and Phylogenetic Analysis

Multiple sequence alignments of protein sequences were performed using MUSCLE [82] with default settings in Ugene 35 software [83] and colored as % of identity. Substitution model tests and phylogenetic analyses were carried out using the MEGA X software [84]. For the maximum likelihood tree [85], the LG substitution model [86] for α-subunit and WAG [85] for the β-subunit was selected assuming an estimated proportion of invariant sites and 4 gamma-distributed rate categories to account for the rate heterogeneity across sites. The gamma shape parameter was estimated directly from the data. The reliability of the internal branch was assessed using the bootstrapping method (1000 bootstrap replicates). The same settings with the JTT substitution model [87] were used for reconstruction with the Neighbor-Joining [88] method, with similar results obtained.

## 5. Conclusions

The functional characterization of Na^+^/K^+^-P-ATPases is still very limited. Our study demonstrated a substantial comparative and evolutionary analysis of plant Na^+^/K^+^-P-ATPases with other taxa and revealed a unique domain architecture and functional differences. Only the α-subunit is ultimately present across prokaryotes and eukaryotes. The absence of animal β- and γ-subunits in plants is very likely related to the matter of the plant way of living, surviving, and anatomy. The potential loss of Na^+^/K^+^-P-ATPases in flowering plants is a strong indicator suggesting escape from saline environments [11,23] and the further employment of more efficient H^+^-P-ATPases for the generation of a steep membrane gradient in low-saline conditions. However, the possibility to discover the P-ATPases with Na^+^ and K^+^/H^+^ activities remains open.

It is still a great challenge to create high-yielding crops with substantially increased tolerance to salinity or any other types of abiotic stresses. Manipulation with membrane transporter genes to improve salt tolerance is one of the most promising approaches to succeed in this. The application of the Na^+^/K^+^-P-ATPases could be presumably an interesting option to achieve a sufficient improvement in salt tolerance [89]. This idea is based on several important issues.

First of all, the recruitment of Na^+^/K^+^-P-ATPases in flowering plants could be efficient to maintain or improve the K^+^/Na^+^ ratio in plant tissue as a key determinant of salt tolerance [90]. Potentially, Na^+^/K^+^-P-ATPases could be a good source of Na^+^ elimination and K^+^ enrichment in plant tissue [11]. Besides this, Na^+^/K^+^-P-ATPases are exceptionally efficient, since the export of three Na^+^ ions and the import of two K^+^ by this enzyme require only one ATP [11]. However, this approach needs to be carefully adapted to vascular plant systems, since these transporters are found mainly in animals and some in green algae.

Secondly, the application of ENA Na^+^-P-ATPases from mosses or fungi could be an alternative option to improve salinity tolerance. However, some studies indicate that the work on this type of ATPases in plants has its price [11]. Therefore, the careful and detailed evaluation of benefits for the heterologous expression approach of Na^+^/K^+^ -P-ATPases or ENA Na^+^-P-ATPases has to be carried out.

Thirdly, it is possible that some putative ouabain-sensitive Na^+^/K^+^-P-ATPases could be identified in flowering plants in the near future [54,55]. Besides this, the discovery of Na^+^/K^+^- transport properties in some other types of plant P-type ATPases could be potentially used for salt tolerance improvement.

Thus, plant Na^+^/K^+^-P-type ATPases are a dynamic and highly adaptive protein family. After the careful analysis of the obtained results, we could suggest that during land colonization ancient Na^+^/K^+^-P-type ATPases changed their ion specificity from Na^+^ to Ca^2+^, which is presented in modern plants. The ion concentrations in terrestrial environments are very different from the saline conditions of sea water. Therefore, it is likely that Na^+^/K^+^-P-type ATPases are no longer required to sustained life on land and could be replaced by more specialized transporters. On the other hand, Ca^2+^ is getting more importance as a messenger and player/part of signaling mechanisms. Additionally, the anatomical structure of the plant has changed, implementing physical barriers against high-salinity terrestrial conditions—suberin, lignins, and lignans. Although some flowering plant species lacking Na^+^-P-type ATPases could thrive in an extremely saline environment, their survival strategy is different to that of Na^+^/K^+^-P-type ATPases transport systems, comprising CPAs, K^+^ channels, PPases, H^+^-P-ATPases, cell wall physical barriers, and specialized organs for excessive salt extrusion from plant tissues (salt bladders and glands, hydathodes).

## Figures and Tables

**Figure 1 plants-10-00221-f001:**
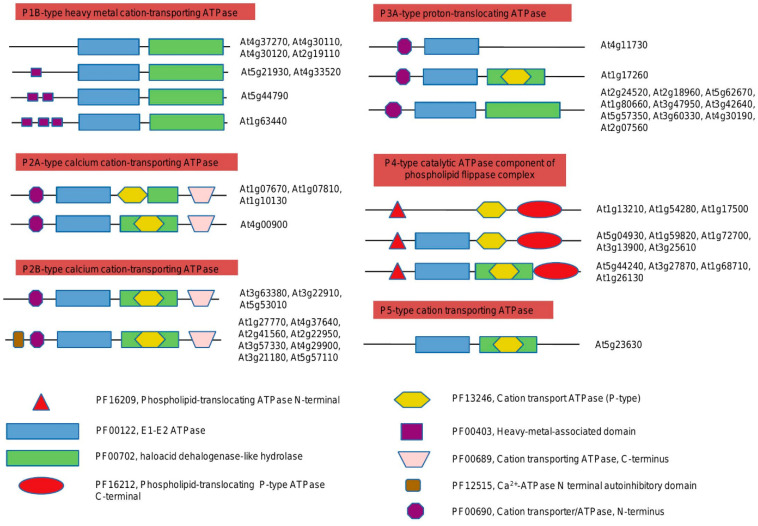
Domain organization of the *Arabidopsis thaliana* P-ATPases. For more details, please see Appendix A.

**Figure 2 plants-10-00221-f002:**
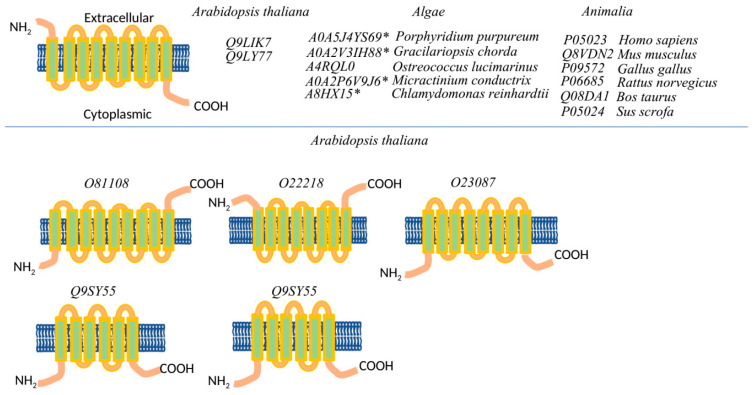
The topology of the transmembrane (TM) domains of the P2B-type ATPases. Comparison of the P2B-type ATPases from *Arabidopsis thaliana*, red and green algae, and Animalia. Proteins encoded with the Uniprot ID. * represents the presence of the β-like subunit insertions in algal species.

**Figure 3 plants-10-00221-f003:**
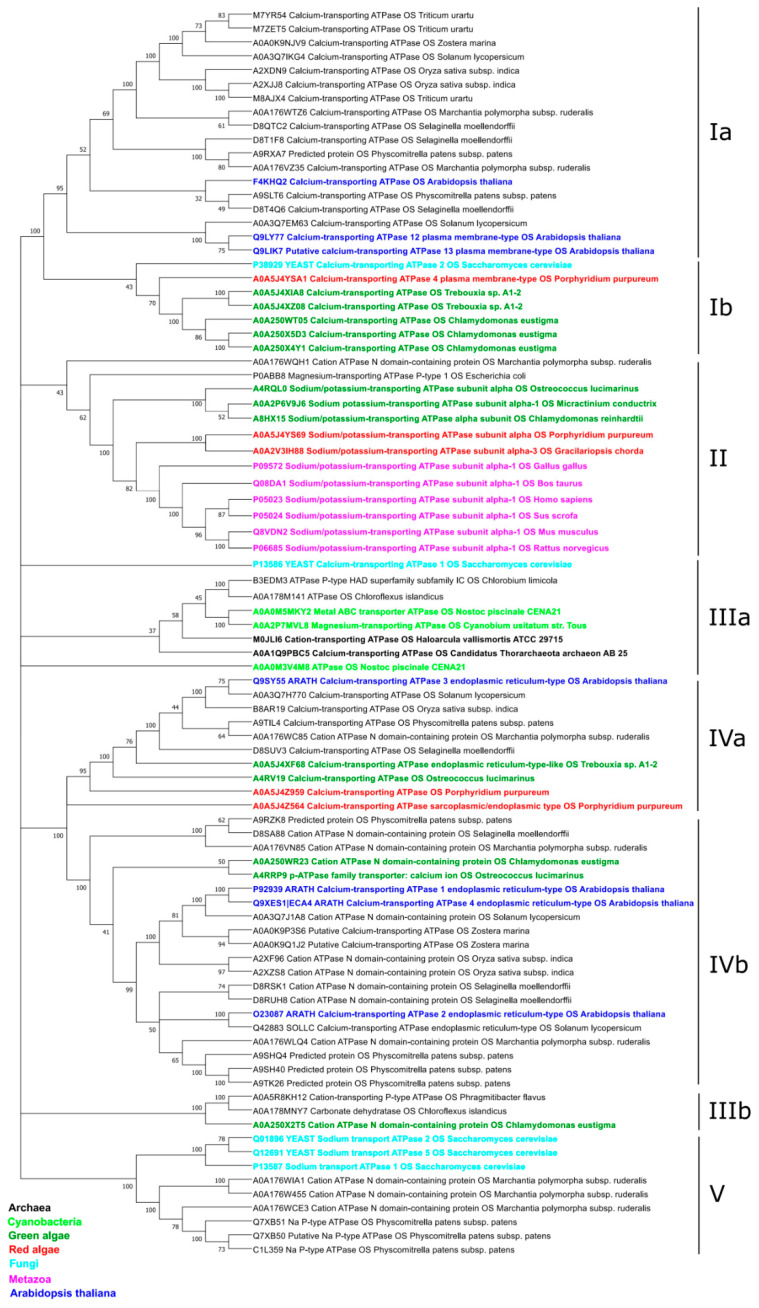
Phylogeny estimation of Na^+^/K^+^-P-ATPases (α-subunit) of the P2B type. The maximum likelihood method and “LG” model were used on 88 sequences with 1000 bootstrap replicates. Phylogeny analysis and substitution model tests were carried out with the MEGAX software. Clusters I–V (including subcategories a and b have been identified based on the branch’s topology.

**Figure 4 plants-10-00221-f004:**
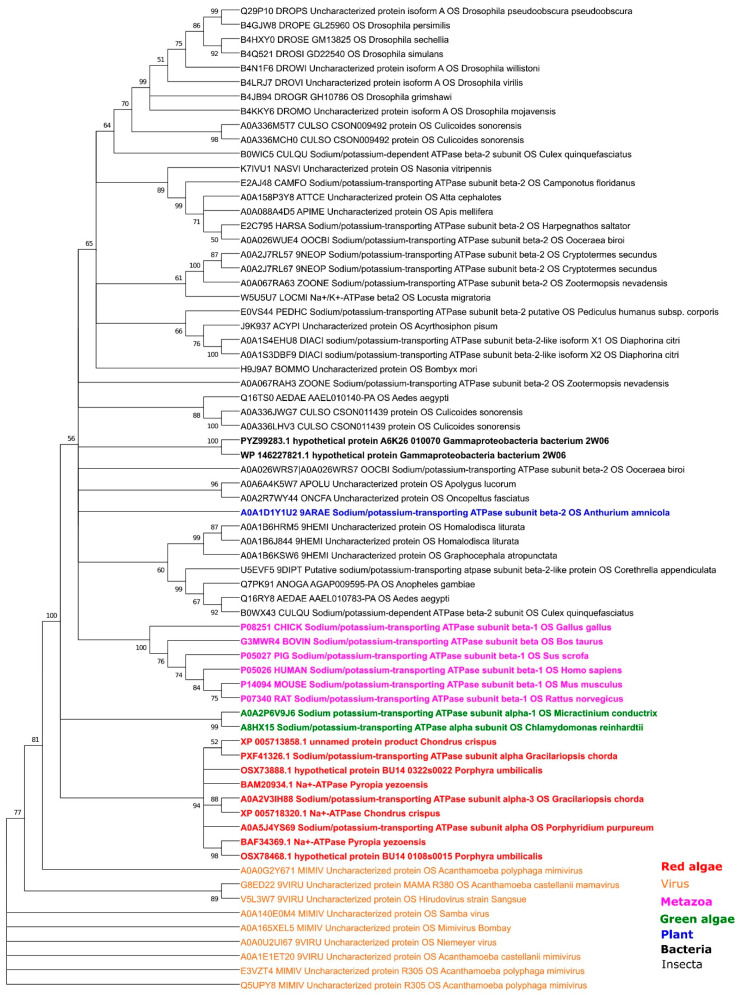
Phylogeny estimation of Na^+^/K^+^-P-ATPases (β-subunit) of the P2B type. The maximum likelihood method and “WAG” model were used with 1000 bootstrap replicates.

**Figure 5 plants-10-00221-f005:**
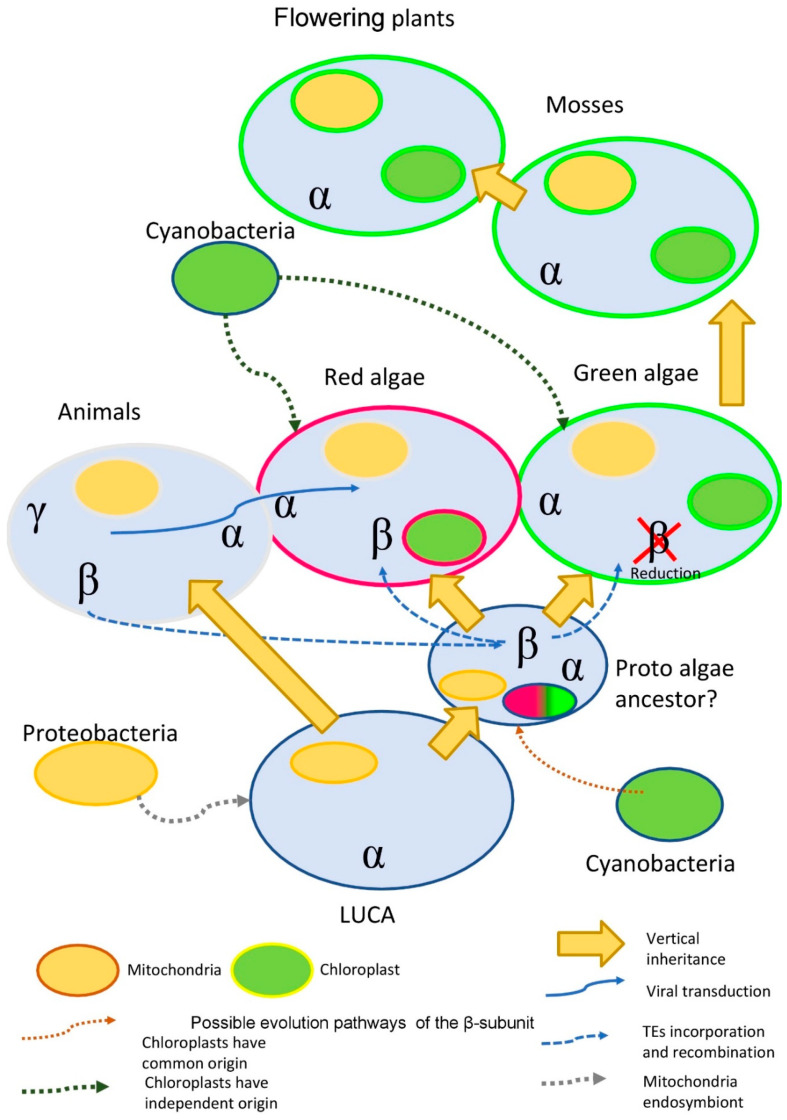
Evolution pathway of the P2B-type ATPases. The incorporation of the ancient proteobacteria (grey dotted line) and cyanobacteria (brown dotted line) gave rise to mitochondria and chloroplasts, respectively. An α subunit as the major P-ATPases form, was, most probably, transferred through the tree of life vertically (wide yellow arrows). Both secondary subunits (β and γ) have been presented only in Animalia. It is possible that the β-subunit was transferred to the proto-algae (before the split on the red and green algae) by the TE (transposable element) elements (dotted blue line), with a further loss in the green algae lineage. Additionally, β could be transferred only to the red algae by viral transduction (solid blue line). Both mechanisms simultaneously are also possible. The black dotted line represents the possibility that red and green algae have acquired cyanobacterial endosymbionts independently.

## Data Availability

The data presented in this study are available on request from the corresponding author.

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
