# Peer review of "Evolution of Plant Na+-P-Type ATPases: From Saline Environments to Land Colonization"

_plants, 2021, doi:10.3390/plants10020221_

Round 1

Reviewer 1 Report

This could be an interesting manuscript after several considerations. Main concerns are the proper background on plant sodium transport and salinity tolerance, number of supplementary figures, the discrepancy between aims, results and conclusions and the clearly of presented figures.

Title and Conclusions: Authors should consider that flowering plants also evolved to the marine environment. In fact they have used the seagrass Zostera marina into the phylogenetic analysis. Low salt environment might not be the unique pressure for the lack of Na+-P-type ATPases in angiosperms.

Lines 42-53. Plant potassium uptake is mediated by different transport systems including a high affinity potassium transporter (HAK5) and channels (AKT and NSCC). In fact, NSCC channels also mediate the uptake of sodium, but as far authors should know HAK5 expression is inhibited by sodium but no evidences have been reported as Na transporter. In this paragraph the term “K+ transporters” is ambiguous and no description about the transport systems used by sodium is reported.

Figure 1 should be reorganized in two columns of three panels. It should contribute to shorten figure size and to clarify the result presentation. Figure legend should include the modeling software.

Figure 3 legend should include some explanation about the number of sequences uses, the categories I to V and the software used to get the phylogenetic tree.

Figure 5. Land plants should be replaced by flowering plants (further review through the text should be done)

Discussion and Conclusions

Despite of discussion seems to be well supported, conclusions are far from the aim of the manuscript. Authors propose a study of molecular evolution of P-ATPases in flowering plants, based on sequence and phylogenetic analysis. Their main findings are not included at the conclusion section. They could mention the potentiality of genome editing tools at the discussion section, not as a conclusion. Conclusions should be based on figure 5. Furthermore, authors could discuss the existence of flowering plants in saline environments, including land and marine environments, these plants lack Na+-P type ATPases but, in fact, can thrive in a extremely saline environment, which points that the Na+-Ptype ATPases might not be a good candidate to improve Na+ tolerance in flowering plants.

Minor:

Review italic for species names

Red and Green Algae appears in capitals sometimes

Line 91: review references numbers one is missing.

Line 94: use between P and ATPase (review all)

Line 136: the absence of, delete “or”…

Line 145: Reconsider results subtitle, comparison to animals?

Number of supplementary figures seems too large. Supplementary figure 1 and 2 could be included in the main text or just quoted into figure 1.

Author Response

Dear Reviewer,

We greatly appreciate your critical evaluation of our manuscript and important suggestions. Our replay (A:) to your comments is given below.

Comments and Suggestions for Authors

This could be an interesting manuscript after several considerations. Main concerns are the proper background on plant sodium transport and salinity tolerance, number of supplementary figures, the discrepancy between aims, results and conclusions and the clearly of presented figures.

Title and Conclusions: Authors should consider that flowering plants also evolved to the marine environment. In fact they have used the seagrass Zostera marina into the phylogenetic analysis. Low salt environment might not be the unique pressure for the lack of Na+-P-type ATPases in angiosperms.

A: Thanks for your remarks. We have intentionally included halophytic or sea species of flowering plants for analysis to reveal the possibility of the existence Na+-P-type ATPases/or other types of P-type ATPases with Na+ transport properties.

Regarding Zostera marina reviewer remarks. We would like to emphasise that Zostera marina evolved from terrestrial ancestors and has colonised back sea environments. Please see ref. Wissler et al 2011. (DOI: 10.1186/1471-2148-11-8).   

Thus, presumably ancestors and land relatives of Zostera marina have already lost the Na+-P-type ATPases. Therefore, we would like to stay on the statement that a low salt environment was one of the key factors to losing Na+-P-type ATPases in angiosperms after land colonisation.    

Lines 42-53. Plant potassium uptake is mediated by different transport systems including a high affinity potassium transporter (HAK5) and channels (AKT and NSCC). In fact, NSCC channels also mediate the uptake of sodium, but as far authors should know HAK5 expression is inhibited by sodium but no evidences have been reported as Na transporter. In this paragraph the term “K+ transporters” is ambiguous and no description about the transport systems used by sodium is reported.

A: We have corrected the term “K+ transporters” and changed it into ion transport systems. Please see L 42.  In fact, K+ transporters belong to several families including CPA, Trk/Ktr/HKT, KT/HAK/KUP. Some members of these transport families could mediate the transport of K+, some will be involved in the transport of Na+, some will contribute to the transport and uptake of both Na+ and K+.  The cation preferences of these transporters will depend on affinity and structural properties.  According to the conducted estimation more than 50 % of overall plant K+ uptake contribution related to K+ channels (Amtmann and Blatt, 2009.  DOI: 10.1111/j.1469-8137.2008.02666.x ). Besides reported K+ transport capability of some Non Selective Cation Channels (NSCC), it is believed that plants have evolved tree different types of K+-channels, Voltage-gated Shaker type channels (AKT, KAT, SCOR, GORK), voltage Independent Two-Pore K+-channels (TPK) and the K+-inward rectifier (Kir) channels.

            The probable role of HAK5 in low-affinity Na+ uptake has been recently reported from reed plants and Arabidopsis. (Wang et al. 2015.   DOI: 10.1007/s10725-014-9964-2; Takahashi et al., 2007. DOI: 10.1007/s00299-007-0364-1)

            Thus, we believe that it cannot be here direct and strict answer about which transport system is involved in K+ or Na+ transport.  

Figure 1 should be reorganized in two columns of three panels. It should contribute to shorten figure size and to clarify the result presentation. Figure legend should include the modeling software.

A: Figure 1 was updated as suggested. No modelling software has been used. Figure 1 was prepared in MS PowerPoint.

Figure 3 legend should include some explanation about the number of sequences uses, the categories I to V and the software used to get the phylogenetic tree.

A: The legend was modified as follow:

Figure 3. Phylogeny estimation of Na+/K+-P-ATPases (α-subunit) of the P2B type. The Maximum Likelihood method and “LG” model were used on 88 sequences; 1000 bootstrap replicates. Phylogeny analysis and substitution models test were carried out with MEGAX software.  Clusters I-V (including sub-categories a and b) have been identified based on the branch’s topology.

Figure 5. Land plants should be replaced by flowering plants (further review through the text should be done)

A: Figure 5 was updated as suggested

Figure 5. Evolution pathway of the P2B type ATPases. The incorporation of the ancient proteobacteria (grey dotted line) and cyanobacteria (brown dotted line) gave rise to the mitochondria and chloroplasts, respectively. An α subunit as the major P-ATPases form, was, most probably, transferred through the tree of life vertically (wide yellow arrows). Both secondary subunits (β and γ) have been presented only in Animalia. It is possible, that β-subunit has been transferred to the proto-algae (before the split on the red and green algae) by the TE elements (dotted blue line), with further loss in the green algae lineage. Also, β could be transferred only to the red algae by viral transduction (solid blue line). Both mechanisms simultaneously are also possible. The Black dotted line represents the possibility, that red and green algae have acquired cyanobacterial endosymbionts independently.

Discussion and Conclusions

Despite of discussion seems to be well supported, conclusions are far from the aim of the manuscript. Authors propose a study of molecular evolution of P-ATPases in flowering plants, based on sequence and phylogenetic analysis. Their main findings are not included at the conclusion section. They could mention the potentiality of genome editing tools at the discussion section, not as a conclusion. Conclusions should be based on figure 5. Furthermore, authors could discuss the existence of flowering plants in saline environments, including land and marine environments, these plants lack Na+-P type ATPases but, in fact, can thrive in a extremely saline environment, which points that the Na+-Ptype ATPases might not be a good candidate to improve Na+ tolerance in flowering plants.

A: Thanks for the suggestions. We have revised our discussion and conclusion. The questions discussed regarding the potentiality of genome editing tools were moved to the discussion section (P12, L 403-416).

In section “Conclusion” we have added one more (last) paragraph dedicated to conclusions based on figure 5 and the lack of Na+-P -type ATPases in flowering plants living in saline environments. (P 15, L 493-506).  

Minor:

Review italic for species names

A: Corrected

Red and Green Algae appears in capitals sometimes

A: Corrected

Line 91: review references numbers one is missing.

A;. Corrected

Line 94: use between P and ATPase (review all)

A: Corrected (L. 83)

Line 136: the absence of, delete “or”…

A: Corrected (L128).

Line 145: Reconsider results subtitle, comparison to animals?

A: Thanks for your suggestion. We have corrected subtitle.

2.1. Domain organisation of the plants P-ATPases, comparison to animals

Number of supplementary figures seems too large. Supplementary figure 1 and 2 could be included in the main text or just quoted into figure 1.

A: We have united supplementary figures S1, S2, S4 and S7, S8 in one supplementary figure1 (S1 - a, S2-b, S4 – c, S7-d, S8-e). Please see supplementary material. As reviewer has suggested we have included a quotation on Supplementary figure 1 in figure 1 legend. 

Reviewer 2 Report

In this study, Authors have tried to study the evolution of P-type ATPases. I have few comments below. 

Please shorten first 4 paragraphs of introduction. 

Line 154 - It should be each, instead of every. Please check out English through out the document. 

Please explain the difference between cation transporting N-terminus and C-terminus, in the domain organization section. 

Please explain softwares used to make these trees in the methods section. 

It is not explained why different number of sequences are taken from different phyla for making these trees. 

Figure 1, Domain organization has not mentioned about alpha and beta subunit. Please explain the missing link. 

Please explain how the presence of cell wall in plants help to negate the requirement of beta subunit. 

Please explain how salt sea-to fresh in land water explain the results from phylogenetic tree. 

Line 319 , “rather stable” could be “conserved”

Line 322, what is AIENENTDE ? Please explain.

Please explain how conservation in Ca2+ binding site and variation in Na+ binding site, is explaining the change in specificity from Na to Ca ? 

Author Response

Dear Reviewer,

We greatly appreciate your time and you remarks.  Our replay (A:) to your comments  is  given below.

In this study, Authors have tried to study the evolution of P-type ATPases. I have few comments below. 

Please shorten first 4 paragraphs of introduction. 
A: We have shortened the first 4 paragraphs of the introduction. Please see the introduction (P1-2, L32-68).   

Line 154 - It should be each, instead of every. Please check out English through out the document. 

A: Corrected (L 146).

Please explain the difference between cation transporting N-terminus and C-terminus, in the domain organization section. 

A: Corrected. Please see L 150-154.   (Those are two conserved N-terminal (PF00690) and C-terminal (PF00689) domains…..)

Please explain softwares used to make these trees in the methods section. 

A: MEGA X software was used (Please, see L 441)

It is not explained why different number of sequences are taken from different phyla for making these trees. 

A: This paper is dedicated to the plants; thus, the majority of species have been taken from plants, other groups have been included for comparison and to highlight the evolution pathway

Figure 1, Domain organization has not mentioned about alpha and beta subunit. Please explain the missing link.

A: Figure 1 is dedicated to the plants, specifically, to Arabidopsis. There are no beta and gamma subunits in Arabidopsis, please see explanation in Introduction (L 110-123), Results (2.1.3) and Discussion (L 267-297, 364-372)

Please explain how the presence of cell wall in plants help to negate the requirement of beta subunit. 

A: beta subunit was suggested to stabilize Na+/K+ transport properties in animals, please, see cited papers [50,51,53]. Furthermore, canonical β and γ-subunits are found only in animal cells. Bacterial as well as plant cells have several members of the P-type ATPases without β and γ-subunits. Therefore, we have suggested that due to different structural cellular properties plant P-type ATPases do not require the presence of β subunits for additional stabilisation of this transport protein.  It is very likely, that due to of different structure of plant cells in comparison with animals – solid cell wall, strengthened by the lignins, suberins, and other compounds, the formation of stress-specific microtubule arrays, inclusion the β and γ subunits in P-type ATPase structure is not necessary. According to the description of animal Na+/K+ -P-type ATPases, the β subunit consists of a small N-terminal cytoplasmic domain; one transmembrane helix; and a large, glycosylated, extracellular domain that covers most of the extracellular surface of the α subunit. Moreover, predicted conserved insert exposed to the extracellular side between M7 and M8 in some algal Na+/K+-P-ATPases suggested to be the potential equivalent of the β-subunit of the animal P-type Na+ pumps. Acquiring the cell wall by plants might lead to loss of extracellular subunits in Na+/K+-P-ATPases and probable structure stabilization due to the formation of a strong cell wall frame for the plant cell.

            However, no direct experimental pieces of evidence are supporting this model. Therefore, we have described it as one of the possible options to explain the missing of β and γ subunits in P-type ATPase in plant cells. We have also included one additional sentence in our discussion indicating about just possibility of the cell wall in stabilizing P-type ATPase structure while stating that no experimental evidence is known so far (L 284- 285).   

            However, to support our point of view the role of the cell wall in salt tolerance is well known and covered in literature: (Engelsdorf et al., 2018, https://doi.org/10.1126/scisignal.aao3070; Feng et al., 2018. /doi.org/10.1016/j.cub.2018.01.023; Kesten et al., 2019, https://doi.org/10.1038/s41467-019-08780-3; Van der Does et al., 2017, https://doi.org/10.1371/journal.pgen.1006832; van Zelm et al., 2020, https://doi.org/10.1146/annurev-arplant-050718-100005)

Please explain how salt sea-to fresh in land water explain the results from phylogenetic tree. 

A: Based on the insertion of the β-subunit-like sequences into the α-subunit mostly in sea green and red algal species, while freshwater species exhibit an absence of such insertion. This clear distinction is visible on our phylogeny tree.

Line 319, “rather stable” could be “conserved”

A: Corrected  (L319)

Line 322, what is AIENENTDE ? Please explain.

A: AIENENTDE – Ca2+-binding site, described in Results section (2.1.1), depicted on Supplementary Figure 1, and Supplementary Figure 6, and further discussed in Discussion section (L 317-336)

Please explain how conservation in Ca2+ binding site and variation in Na+ binding site, is explaining the change in specificity from Na to Ca ? 

A: Na+-binding sites are missing in plants’ analyzed P-type ATPases, Ca2+-binding, from the other side, is rather conserved and omnipresent in all analyzed species. This suggests that Ca2+ plays a role in physiology, while Na+ - is rather an environmental stress-agent, required a special transport system and deactivation.

Reviewer 3 Report

The manuscript provide a comparative and evolutionary analysis of plant Na+/K+-P-ATPases with other taxa. Results revealed unique domain architecture and functional differences useful to explain (or at least partly support)  evolutionary aspects of land plants management of salt stress during terrestrialization.

I found the manuscript very interesting and well prepared; the bibliography is adequate and accurate. Results presentation is clear and the discussion coherent with results.

Author Response

Dear Reviewer,

We greatly appreciate your time and high remark of our manuscript.

The manuscript provide a comparative and evolutionary analysis of plant Na+/K+-P-ATPases with other taxa. Results revealed unique domain architecture and functional differences useful to explain (or at least partly support)  evolutionary aspects of land plants management of salt stress during terrestrialization.

I found the manuscript very interesting and well prepared; the bibliography is adequate and accurate. Results presentation is clear and the discussion coherent with results.

A: Thank you very much for your comments

Round 2

Reviewer 1 Report

Minor changes: 

Abstract, lines 13 to 16. There is a contradiction between: Na+/K+-P-ATPases are the "omnipresent" family of pumps...  and Na+/K+-P-ATPases are considered "to be absent" in flowering plants. Please replace "omnipresent" by "main" in animal cells for example. See line 49-50.

Line 101-102, correct the position of algae species, cite Chondrus crispus (red algae) firstly.  

Author Response

Dear Reviewer,

We greatly appreciate your remarks and careful reading of our manuscript. We have corrected all remarks suggested by the reviewer.

Our replay (A:) to your comments is given below.

Minor changes: 

Abstract, lines 13 to 16. There is a contradiction between: Na+/K+-P-ATPases are the "omnipresent" family of pumps...  and Na+/K+-P-ATPases are considered "to be absent" in flowering plants. Please replace "omnipresent" by "main" in animal cells for example. See line 49-50.

A: We have corrected this sentence. We have exchanged the word “omnipresent” to “widely distributed among the different taxa family of pumps….” (L. 14-16)

Line 101-102, correct the position of algae species, cite (Chondrus crispus red algae) firstly.  

A: We have exchanged the position of red algae Chondrus crispus. Now we  site  Chondrus crispus as the first algal species in the sentence (L. 102-103)

This manuscript is a resubmission of an earlier submission. The following is a list of the peer review reports and author responses from that submission.

Round 1

Reviewer 1 Report

The manuscript entitled “Evolution of Plant Na+- P-type ATPases: from saline environments to the land colonization” by Dabravolski and Isayenkov investigated the evolution of Na+ P-type ATPase based on in silico analysis. The authors first analyzed the domain architecture of Arabidopsis P1 to P5 type ATPase based on sequence and domain alignment. Then they compared the topological transmembrane domains of P2B type ATPases in Arabidopsis and other organisms. The authors next generate phylogenic trees of P2 type ATPase α and β subunits and found that the α and β subunits are close to the ones in red/green algae. Finally, they propose an evolutionary path of P2B type ATPase from LUCA to animals and plants. Overall, this manuscript tries to understand a very interesting evolutionary path of ATPases and attempt to answer why vascular plants lack Na+/K+-ATPases. I had difficulties reading this manuscript as the language is very poor but, after coming through it, I cannot agree with most conclusions and do not think this research show strong novelty and originality. What is most critical, the work contains multiple flaws.

Major points:

1) Figure 1: The depicted domain organization of the Arabidopsis thaliana ATPases has not been subject to rigorous quality control and is therefore meaningless. It is most doubtful that the suggested variation in P-type ATPase signature motifs is valid. In pfam searches, all matches scoring above the curated threshold, are aligned back to the profile to generate the full alignment. The threshold is chosen to avoid the inclusion of false positives but will also result in false negatives. For example, from Figure 1, it appears that three P4 ATPases in Arabidopsis (At1g13210, At1g54280 and At1g17500) do not have the pfam motif PF0122. What does this mean? PF0122 covers the A domain of P-type ATPases, which includes a glutamate residue, which is essential for dephosphorylation of the E2P phosphoenzyme. P4 ATPases have a somewhat divergent A domain in which for example the glutamate is found in a DGET motif and not in the motif TGES, which is typical for most other P-type ATPases. Inspection of the A domain of the three excluded Arabidopsis P4 ATPases reveal that they indeed have the DGET motif as well as other conserved A domain motifs. Thus, the threshold value has been set so high that the automatic pfam annotation as a result has listed these true positives as false positives.

2) Figure 2: According to the legend, all proteins in the figure are P2B type ATPases but this is wrong as it also includes Na+/K+ ATPases. According to common terminal terminology, autoinhibited Ca2+-ATPases are defined as P2B ATPases whereas Na+/K+-ATPases are P2C ATPases.

3) Figure 2: The indicated topology of transmembrane domains does not make sense. From the figure it appears that P2 ATPases can have their terminal domains in different orientations with respect to the membrane. Not a single structure of P2 ATPases supports this assumption. All structures and experimental evidence point to the N-terminal and the C-terminal domains being both exposed to the cytoplasm.

4) Phylogenetic analyses in Figures 3 and 4: It is not correct that the trees show P2B ATPases. In additioin to P2B sequences, the trees include sequences representing P2C (Na+/K+) and P2D (Na+ or K+) ATPases. Bootstrap values for major branches are very low, and therefore the authors should refrain from use them as a basis for strong conclusion such as using them as a basis for the subdivision into named major clades. I would only accept a value below 70 if supported strongly by Bayesian inference analysis.

5) l. 187: “All identified Na+/K+ ATPases have been located to the clade II. Interestingly, this clade includes evolutionally very distant species: animals, red and green algae. Bacterial magnesium-transporting ATP (Escherichia coli P0ABB8) and liverwort cation ATPase (Marchantia polymorpha A0A176WQH1) have served as parental forms.”; There is absolutely no basis for this claim. It is based on extremely low bootstrap support values.

6) Figure 4: Sequences have not been subject to rigorous quality control, which makes the phylogenetic analysis meaningless: The sole beta-subunit sequences from prokaryotes (both from Gammaproteobacteria bacterium 2W06) are likely contaminating sequences as they are almost identical to sequences from the prawn Penaeus vannamei (XP_027217779.1 and AEE25938.1). The authors have to modify their statement that beta-subunits are present in prokaryotes. The sole beta-subunit sequence from land plants (the araceae Anthurium amnicola) is most likely an error. When I do Blast-searches, I do not find any beta-subunit-like sequence in Anthurium amnicola, and not hits are received when I search in several databases for the indicated accession numbers (A0A1D1Y1U2 and 9ARAE). The authors have to modify their statement that beta-subunits are present in land plants.

7) Figure 5: This figure is meaningless. What has evolution of Na+/K+-ATPases to do with endosymbiosis of mitochondria and chloroplasts? Blue dotted lines seem to indicate that the beta-subunit has passed from animals to plants (algae) by horizontal gene transfer. There is absolutely no evidence for that.

8) There are multiple mistakes in grammar throughout the whole manuscript, e.g. in the line 13, “Na+/K+-ATPases is” should be “Na+/K+-ATPases are”; in the lines 15 and 16, “is” and “attempt” should be consistent in terms of singular vs. plural; in line 50, the first “primary” is redundant in the sentence “are considered to be primary the primary Na+ efflux systems in plant cells”.

Other comments:

9) The Abstract states: “We were capable to trace gradual site-replacement starting from cyanobacteria to the modern land plants.” However, this part of data is not clearly showed.

10) There is no definition of ‘plants’ in the manuscript. ‘Algae’ is not a taxonomic group but Chlorophyta (green algae) and Streptophyta are included in Viridiplantae. A critical discussion whether algal Na+/K+-ATPase-like sequences encode functional Na+/K+-pumps is lacking.

11) NaCl can result in plant stress as a result of Na+ cytotoxicity but its role in causing water stress is barely mentioned.

12) l. 56: There is no large protein family called ‘ATPases’. Multiple protein families comprise ATPases. The authors refer to P-type ATPases.

13) l. 158: There are not several Na+/K+-ATPase structures but two.

14) l. 165: Coordinating residues coordinating what ion(s)? Arabidopsis pumps do not coordinate neither Na+ nor K+.

15) l. 171: Why is this interesting?

16) l. 67: P5 ATPases are likely transmembrane helix dislocases.

17) The enzyme names are not consistent: P-type ATPases and P-ATPases, Na+/K+-ATPases and Na+/K+ ATPases.

18) Line 31 “…productivity salinity…” should be “…productivity by salinity…”

19) Line 33, reference #2 should be more specific, official files should also be updated with the latest version (http://www.fao.org/global-soil-partnership/areas-of-work/soil-salinity/en/).

20) Line 41, the sentence “K+ and Na+ are competitors” needs to be more specific to explain in which aspect they are competitors for better readership.

21) Line 59, the functions of P1 to P5 type ATPase could be better explained about what are already known for these types.

22) “Higher plants”, “vascular plants” and “flowering plants” are all used in the manuscript, it is better to unify the name for less confusion.

23) Line 85, 86 and 87, P. patens should be italic. Please check the similar problems and correct them.

24) Line 102-104, the description should tone down.

25) Line 113 and 123, “lucking” should be “lacking”.

26) Figure 1, it’s better to provide explanation why AT3G63380/AT3G22910/AT5G53010 have the same domain architecture as P2A type? Also, Figure 1 needs a legend.

27) Line 153, it is not clear what is “this approach”.

28) Tables are missing.

29) In Figure 4 legend, Arabidopsis is missing (dark blue, bold).

30) The whole manuscript requires substantial correction and rewriting to avoid confusion and improve readability.

Reviewer 2 Report

The manuscript by Dabravolski and Isayenkov is difficult to understand in the present version. The objective of this work is not clear. Phylogenetic analysis of P-type ATPases has been performed by other authors (Palmgren, Rodríguez-Navarro, among others). The novelty of this work is not clear when compared to what is already known. Authors show a detailed phylogenetic analysis of different P-type ATPases, but the results and discussion of data are not clear and the manuscript would need extensive rewriting. A comparison between P-type ATPases, i.e Na+/K+ ATPases, Ca2+-ATPases  and  H+-ATPases of different organisms from an evolutionary point of view could be interesting for publication in Plants. However, if this is the objective of this manuscript, such a comparison is not presented in an understandable way in the present version.

Na+/K+ ATPases and Na+ pumps have not been described in higher plants. Through the manuscript, authors refer many times to plant Na+/K+ ATPases , but they have only been found in some algae. Na+ ATPases have only been found in mosses.  Authors should refer to primitive plants (algae and mosses) when mentioning Na+/K+ ATPases  or Na+ ATPases , since the term plants also includes higher plants.

Finally, the use of the English language should be improved for a better understanding.
